# Skin muscle is the initial site of viral replication for arboviral bunyavirus infection

Christine A. Schneider[1], Jacqueline M. Leung[2],
Paola Carolina Valenzuela-Leon[3], Natalya A. Golviznina[4], Erik A. Toso[4],
Darko Bosnakovski[4], Michael Kyba[4], Eric Calvo[3] & Karin E. Peterson[1]✉

The first step in disease pathogenesis for arboviruses is the establishment of infection following vector transmission. For La Crosse virus (LACV), the leading cause of pediatric arboviral encephalitis in North America, and other orthobunyaviruses, the initial course of infection in the skin is not well understood. Using an intradermal (ID) model of LACV infection in mice, we find that the virus infects and replicates nearly exclusively within skin-associated muscle cells of the panniculus carnosus (PC) and not in epidermal or dermal cells like most other arbovirus families. LACV is widely myotropic, infecting distal muscle cells of the peritoneum and heart, with limited infection of draining lymph nodes. Surprisingly, muscle cells are resistant to virus-induced cell death, with long term low levels of virus release progressing through the Golgi apparatus. Thus, skin muscle may be a key cell type for the initial infection and spread of arboviral orthobunyaviruses.

La Crosse virus (LACV), a member of the California Serogroup of Orthobunyaviruses, is the primary cause of pediatric arboviral encephalitis in North America. One of the first key steps in arboviral infection is deposition of the virus by the mosquito during blood meal acquisition. Virus can be deposited in the epidermis and dermis of the skin while the mosquito actively probes to locate a capillary to initiate feeding. However, very little is understood about what occurs during the initial steps following deposition of LACV in the skin, including what cells are infected. Indeed, there are limited studies on orthobunyavirus infection following either mosquito feeding or intradermal (ID) inoculation in general. Beaty et al. infected mice with LACV by subcutaneous (SQ) injection and detected viral antigen in the skin in a portion of the symptomatic weanling mice[1]. However, virus was found in striated muscle, fat, vascular tissue and other dermal tissues which could be a result of the disseminated viremia that would be expected from this type of disease model[1]. With lower, non-lethal doses of virus, antigen was only detected in the skin for a few days post infection, suggesting the virus can also be cleared over time[1]. The most common route used for LACV inoculation in mice is intraperitoneal injection

(IP). Analysis of peripheral tissue following this route found infection of skeletal muscle, heart, spleen, and several other organs[2], although individual tissue infection was highly varied between mice and was only consistent in lymph nodes (LN). Thus, a better understanding of LACV and other bunyavirus infection of the periphery is needed, particularly using an appropriate model that mimics delivery by a mosquito into the skin.

Studies with other neuroinvasive arboviruses show preferences for dermis and epidermis resident cells. West Nile virus (WNV) infects both keratinocytes and Langerhans cells in the skin while Venezuelan Equine Encephalitis Virus (VEEV) targets Langerhans cells, macrophages, and dendritic cells[3–6]. For both WNV and VEEV, dissemination and eventual central nervous system (CNS) infection is accentuated by transport of virus from the skin to distal sites by migratory immune cells[4–6]. This phenotype is not restricted to neurovirulent arboviruses. The two most common arboviruses, Dengue Virus (DENV) and Chikungunya Virus (CHIKV), also readily infect the dermis and epidermis after ID infection, with a bias towards migratory immune cells and fibroblasts in human and mouse skin[7–10]. Both viruses can subsequently

[1]Laboratory of Neurological Infections and Immunity, Rocky Mountain Laboratories, National Institute of Allergy and Infectious Diseases, National Institutes of Health, Hamilton, MT, USA. [2]Research Technologies Branch, Rocky Mountain Laboratories, National Institute of Allergy and Infectious Diseases, National Institutes of Health, Hamilton, MT, USA. [3]Laboratory of Malaria and Vector Research, National Institute of Allergy and Infectious Diseases, National Institutes of Health, Rockville, MD, USA. [4]Lillehei Heart Institute, University of Minnesota, Minneapolis, MN, USA. ✉e-mail: petersonka@niaid.nih.gov

be found within the draining LN from the site of infection, eventually leading to hematogenous spread and dissemination to sensitive tissues. Direct infection and destruction of skeletal muscle cells precipitates much of the arthritis symptoms described by patients with CHIKV and has been suggested as a cause of DENV and CHIKV myocarditis[11–13]. Thus, many arboviruses infect keratinocytes, fibroblasts, and immune cells in the skin and migration of infected immune cells leads to virus spread throughout the host.

For disease modeling, the site of infection and method of delivery can be important considerations that alter the relevant cell types for dissemination and the course of disease. Some studies designed to mimic skin infection are performed with sub-cutaneous injection which would place the virus below the skin and deeper than would be expected from mosquito injection. ID inoculation is the closest model that mimics the natural human infection by mosquito. We therefore chose to investigate if the skin is productively infected following an ID injection of LACV and investigated the pattern of dissemination from the skin to the brain as a step towards modeling vector-borne disease.

Here, we show that LACV infection is largely restricted to skin-associated skeletal muscle cells with a marked absence of infection in other dermal or epidermal cells. The virus is widely myotropic but does not disseminate through the LN. Further analysis of infected skeletal muscle cells using an ex vivo system showed that muscle cells have low levels of virus release and do not readily undergo apoptosis. Thus, skin muscle cells may be an important site for virus infection and prolonged replication in the skin.

## Results

### LACV infects skin-associated muscle after intradermal infection
Mammalian hosts acquire LACV when an infected mosquito delivers virus directly into the skin during feeding. Previous work with LACV and related orthobunyaviruses have modeled this by SQ injection and noted a non-specific pattern of virus infection in skin cells likely due to hematogenous spread[1]. To directly examine LACV infection and cell tropism in the skin, we infected weanling mice by ID injection with $10^5$ plaque forming units (PFU). This virus dose results in lethal neurological disease in weanlings and is within the range of infectious virus found within LACV infected mosquitoes[14]. We utilized C57BL/6 wild-type (WT) mice, as well as C57BL/6 mice that were heterozygous for $Cx3cr1^{GFP}$ and $Ccr2^{RFP}$ ($Cx3cr1^{+/GFP}$ x $Ccr2^{+/RFP}$) to allow for easy detection of myeloid cells. Skin tissue was removed at 48 h post infection (hpi)/ 2 days post inoculation (dpi) and examined for virus by fluorescence microscopy. Interestingly, we found that LACV was almost exclusively located within the myosin-expressing muscle cells of the panniculus carnosus (PC), a thin, striated muscle layer underlying the dorsal and ventral skin on the mouse thorax and abdomen (Fig. 1A). We observed no staining with LACV antibody present in any location in mock-infected mice (Fig. S1B). As arbovirus infection can be influenced by vector saliva, we also injected LACV that was pre-incubated with salivary gland extract (SGE) isolated from the *Aedes triseriatus* mosquito, the primary vector for LACV. SGE did not alter LACV cell tropism, with the overwhelming majority of the infected cells still localized to the muscle cells (Fig. 1B). In $Cx3cr1^{+/GFP}$ x $Ccr2^{+/RFP}$ mice, the CX3CR1-GFP cells within the dermis did not stain positive for LACV in the absence or presence of SGE, ruling out infection within macrophages or keratinocytes (Fig. 1A). In WT mice, we stained for CD90 (Thy-1), an antigen widely expressed in immune cells, fibroblasts, epidermal cells, and neuronal cells and found no LACV staining within those cells (Fig. 1B). We quantified the number of infected cells compared to the total number of muscle cells across 2–4 sections from 4 individual mice infected with LACV alone and found that an average of $10.6 ± 4.9\%$ of muscle cells were infected per section (Fig. 1C).

To determine if infection of PC muscle cells was a unique finding for LACV or a potential result of route of inoculation, we injected weanling mice with $1 × 10^5$ PFU West Nile virus (WNV; 385-99 strain).

Virus was inoculated with and without SGE from *Aedes aegypti* mosquitos, following the same protocol as above. In contrast to LACV, WNV was found throughout the dermis independent of the presence of SGE, including within cells overlying the muscle cells, but was absent from all myosin positive PC cells (Fig. S2). Thus, LACV and WNV infected different cell types within the skin, despite identical methods of inoculation.

LACV-infected PC muscle cells in the skin cross sections were not adjacent to one another, as is expected with foci formed during virus infection. However, mature skeletal muscle cells are long, multi-nucleated myofibers. We therefore hypothesized that these "single" cell infections might indicate infection throughout the length of the myofiber. To investigate this, we removed the entire region of the skin surrounding the injection site from mice infected ID with LACV and performed immunostaining and optical clearing. Confocal imaging confirmed that infected, intact muscle fibers contain virus throughout their length, with the virus primarily localized as puncta surrounding the myosin-actin bundle within the myofiber (Fig. 1D). We next asked if LACV replicates within PC muscle cells. RNA probes designed to detect LACV genomic RNA (gRNA), and the replicative intermediate RNA (rRNA) were used to analyze serial sections of injection site skin at 24 hpi (1 dpi). Both gRNA and rRNA probes were detected in cells below the hypodermis with morphology suggestive of being within the muscle layer (Fig. 1E) compared to a lack of probe staining in mock infected mice (Fig. S1A). Together, these data indicate that PC muscle cells are the primary site of LACV infection and are capable of supporting replication within the skin.

### LACV infects muscle cells in other tissues at later stages of virus infection
Next, we tracked the kinetics of viral dissemination. The injection site skin was removed at time points ranging from 0–4 dpi (0–96 hpi) and analyzed for infectious virus by plaque assay. After an initial drop in infectious virus from the injection at $T = 0$ dpi, virus production increased over the next 3 days, indicating virus replication was occurring within the skin (Fig. 2A). Virus was detectable in the ipsilateral thigh muscle inferior to the infection site at the initial times of inoculation, indicating some inoculum may pass through the skin into the subdermal tissues after the injection and replicates there in some mice (Fig. 2B). Interestingly, the level of virus in the ipsilateral thigh muscle (ITM) did not reach the same level of infection as the skin and was highly variable (Fig. 2B). Virus in the plasma was consistently observed by 2–4 dpi (Fig. 2C) indicating the LACV infection of the PC muscle following ID infection resulted in viremia.

For many arboviral diseases, virus is trafficked through the LN before viremia and widespread dissemination occurs[10,15–18]. To assess if this was true in our model, we used qRT-PCR for increased sensitivity and measured viral RNA in the inguinal LN located directly adjacent to the skin injection site. Compared to the skin, LACV transcripts within the LN was only observed in half of the animals at any given time point and did not increase over time (Fig. 2D). This suggests that, although virus replication is ongoing in the skin and viremia can be established, the LN does not appear to be a key site of virus proliferation.

Previous work by Janssen et al. used a model of SQ inoculation of LACV in highly susceptible newborn (suckling) mice and found viral antigen in striated skeletal muscle at later times of infection[19]. We hypothesized that, given the infection of the PC muscle in the skin, LACV might be widely myotropic during the dissemination phase. Indeed, viral transcripts were detectable in the ipsilateral thigh muscle in all samples within 10 min of the injection and remained detectable in most animals over 4 dpi (Fig. 2E). Analysis of muscle tissues more distal to the inoculation site, including the peritoneal wall, heart, and contralateral thigh revealed viral transcripts in peritoneal muscle adjacent to the skin injection site in 4 of 5 animals and in the heart of all 5 of 5 animals by 4 dpi (Fig. 2E). Thus, multiple muscle tissues were infected

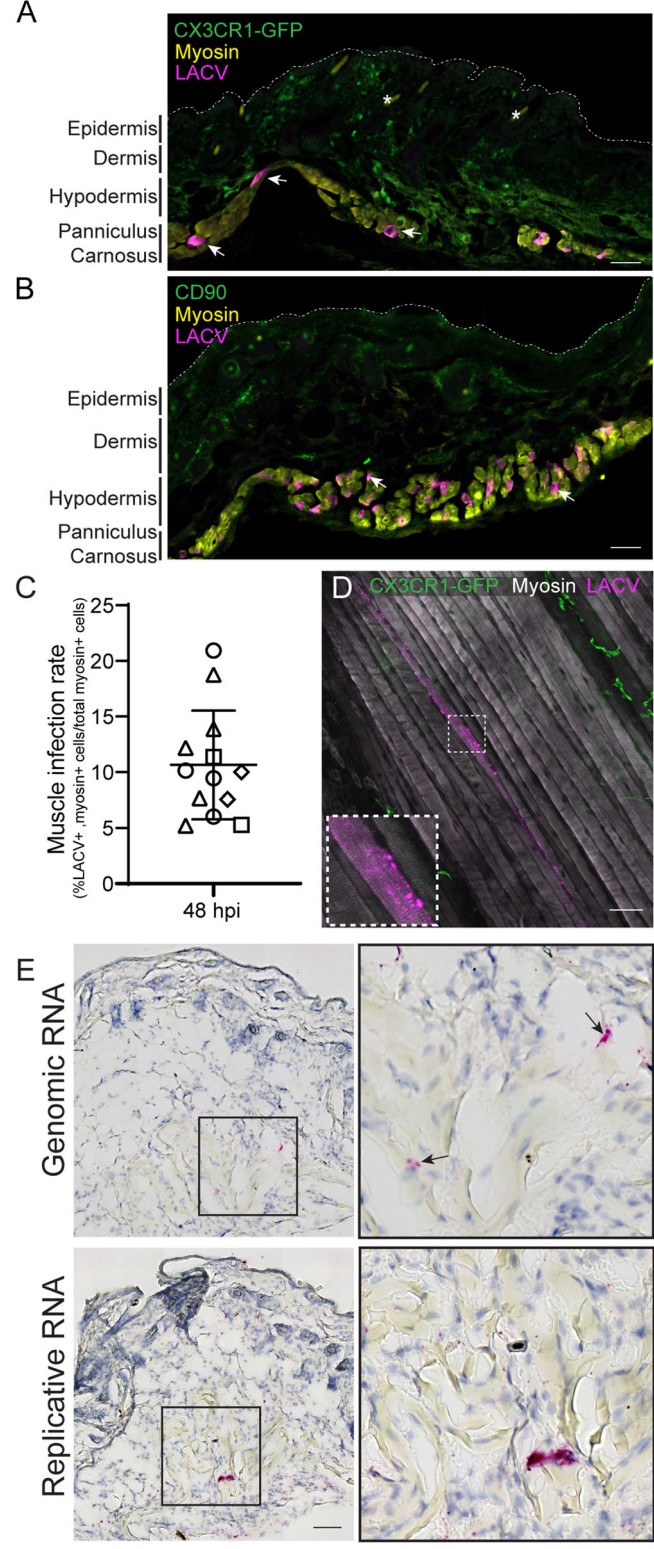

**Fig. 1 | LACV infects and replicates in skin-associated muscle.** Mice were infected with LACV and back skin was removed at 1 dpi (**E**) and 2 dpi (**A, B, D**). For (**B**), LACV was pre-incubated with mosquito salivary gland extract before injection. Skin sections (**A, B**) or cleared whole skin pieces (**D**) were stained with antibodies against LACV (magenta) and GFP or CD90 (green) (**A, B**) and anti-myosin (yellow (**B**), white (**D**)). Infection rate in the skin was quantified per skin section as a percent of LACV+, myosin+ cells from total myosin+ cells and plotted as individual points per field of view with different symbol shapes representing four separate mice (**C**). Error bars represent mean ± SD. Arrows indicate infected cells and asterisks denote autofluorescent hair follicles. Dotted line in (**A, B**) mark outer boundary of the epidermis. Sections were also assayed by colorimetric in situ hybridization for LACV genomic RNA or LACV replicative intermediate RNA (**E**). Insets are 3x zoom from original images. Nuclei were visualized by hematoxylin staining (**E**). Microscopy is representative of 2 (**D**) or 4 (**A, B, E**) mice per analysis. Scale bars represent 50 μm. Source data are provided as a Source Data file.

staining relative to total myosin staining produced a similar ratio between two independent mice across 3–4 sections per mouse (Fig. S3A) suggesting consistency in the level of muscle infection between sections and mice.

Virus was also found in contralateral thigh muscle, although this was greatly reduced and at a much lower incidence than the ITM (Fig. 2E). Consistent with the spread of virus to the CNS within 5 days, we found LACV transcripts in the brain of all mice by 4 dpi (Fig. 2E). Taken together, these results show LACV infects and replicates within the PC muscle for at least 4 dpi and LACV is widely myotropic before the onset of neurological disease.

## Muscle cells are infected in a route independent and species independent manner

In mice, the PC is found predominantly on the midsection and does not cover the entire animal[20]. There are, however, thin striated muscle layers associated with the skin in other locations such as the ear pinna, which contains rarefied, thin bands of auricular muscle throughout the ear[21]. To evaluate if LACV infection in the skin is restricted to the PC muscle, we infected mice in the ear pinna and evaluated which cells were virus positive at 2 dpi, comparing these tissues with mice infected on the back skin. We found LACV within myosin positive muscle cells in the ear pinna similar to that found in the PC (Fig. 3A, C). In ear sections lacking substantial bands of auricular muscle, no virus was detected in any cells (Fig. 3B), reinforcing that LACV was preferentially located within muscle cells. Despite having less muscle coverage in the ear pinna skin compared to the back skin, mice inoculated in the ear pinna developed neurological disease, with no significant difference in the kinetics of disease onset compared to ID inoculation (Fig. 3D). Taken together, these data suggest that LACV will infect skin-associated muscle independently of location and that even infection of tissues with small areas of skin-associated muscle can lead to virus infection, dissemination, and neurological disease.

We next analyzed whether IP inoculation resulted in muscle cell infection. The closest striated skeletal muscle associated with IP inoculation would be the muscle in the peritoneal wall. IHC analysis of peritoneal muscle from IP inoculated mice showed strong positive staining for LACV (Fig. 3E). Thus, all three routes of peripheral inoculation, ID in the back, ID in the ear pinna, and IP resulted in LACV infection of nearby skeletal muscle tissue.

Lastly, we asked if human muscle cells can be infected with LACV. We infected immortalized human myoblasts originally derived from a healthy human donor with LACV. By 7 dpi, numerous infected muscle cells were detected by immunostaining (Fig. 3F). Although the immortalized myoblasts did positively stain for myosin (Fig. 3F), we did not differentiate them, so they did not show the sarcomere development characteristic of mature myotubes. Thus, muscle cells may be a key cell type for peripheral virus replication of LACV

with LACV within the first 2 days, prior to the detection of virus in the CNS (Fig. 2E).

To confirm the viral transcripts detected in the heart represented skeletal muscle cell infection and not adherence to the interior surface of ventricles or atria, we performed immunohistochemistry at 6 dpi and identified numerous LACV-positive cardiac muscle cells, with a particular concentration within the left atria in two independent mice (Fig. S3B) which was confirmed to not be the result of non-specific binding of secondary antibodies (Fig. S3B). Quantification of LACV

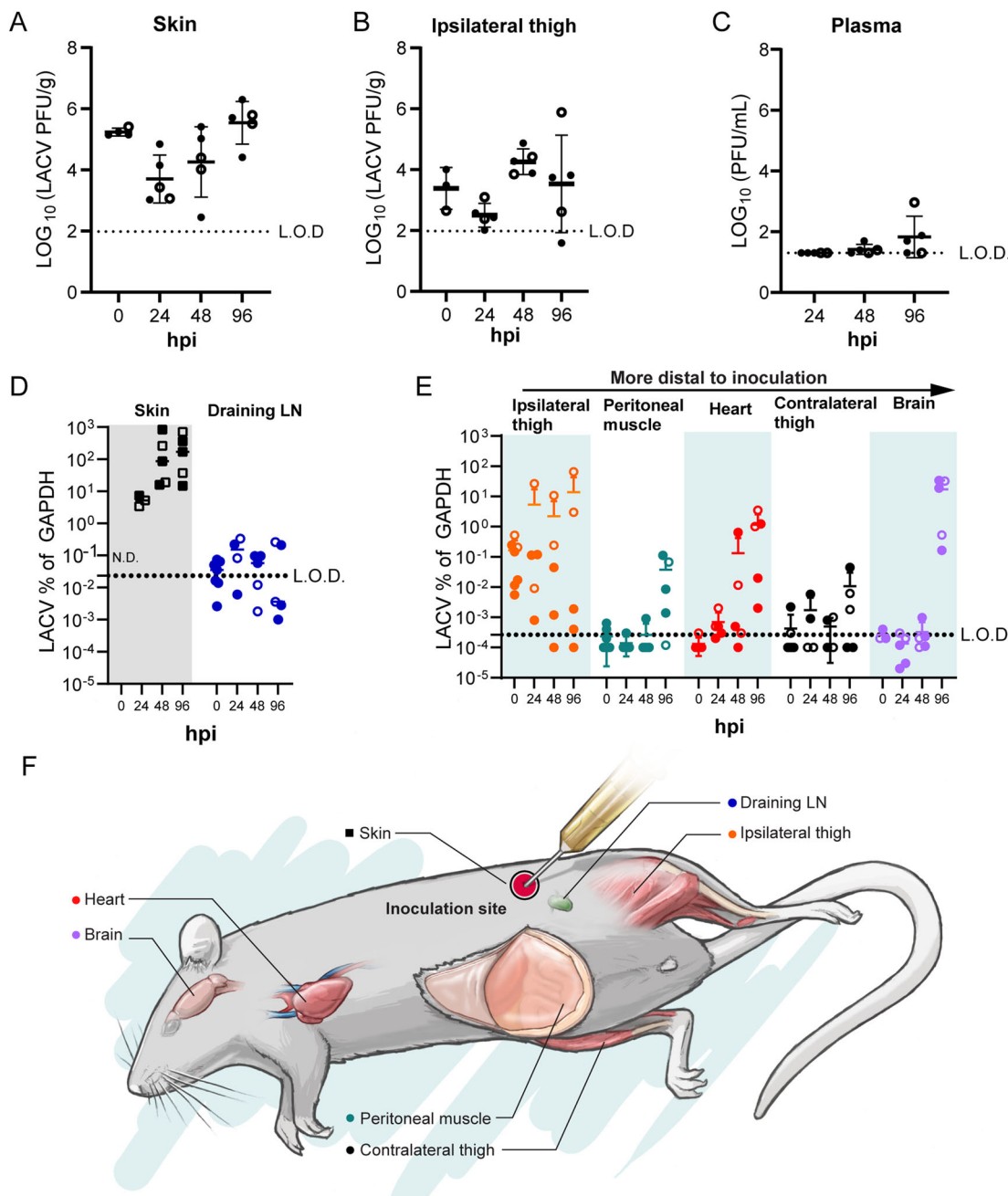

**Fig. 2 | LACV infects the skin, skeletal muscle, and brain without trafficking through the LN.** Weanling mice were infected ID with LACV and tissues harvested for plaque forming assay (**A**–**C**) and qPCR for LACV S segment normalized to housekeeping gene GAPDH (**D**). Each symbol indicates an individual mouse (biological replicate) for each time point. L.O.D limit of detection calculated from average of mock infected mouse tissues. N.D. not done as entire injection site was used for plaque assay. Time 0 hpi are tissues from infected mice isolated within 10 min post injection. Data graphed as mean with error bars representing SD. Filled symbols represent males and open symbols represent females. Shading (**D**, **E**) added to delineate different tissues. **F** Drawing provides a guide to where each tissue was taken for analysis from the mouse. Source data are provided as a Source Data file.

independent of initial inoculation site or species. Surprisingly, infected human myoblasts showed only limited cell death following LACV infection as measured by cell viability, even when infected with a very high multiplicity of infection (MOI) (Fig. 3G). A decrease in survival was observed at 7 dpi, but this was not significant compared to mock controls. This contrasts to previous studies of LACV infection of neurons, the primary cell type infected in the brain, which have significant cell death as early as 1 dpi in vitro, with only few cells remaining by 3 dpi[2,22,23].

## LACV infection does not induce muscle cell death in vivo or ex vivo

LACV infection of neurons induces apoptosis through a caspase-3 dependent mechanism, with infected cells showing strong active caspase-3 staining[22,24] (Fig. 4A). To determine if LACV also induces death of muscle cells in vivo, we stained sections from brain, skin, and muscle with antibodies that detect cleavage of the pro-apoptotic protein caspase-3. We observed active caspase-3 staining in the brain (Fig. 4A first row, yellow co-localization staining as well as lower

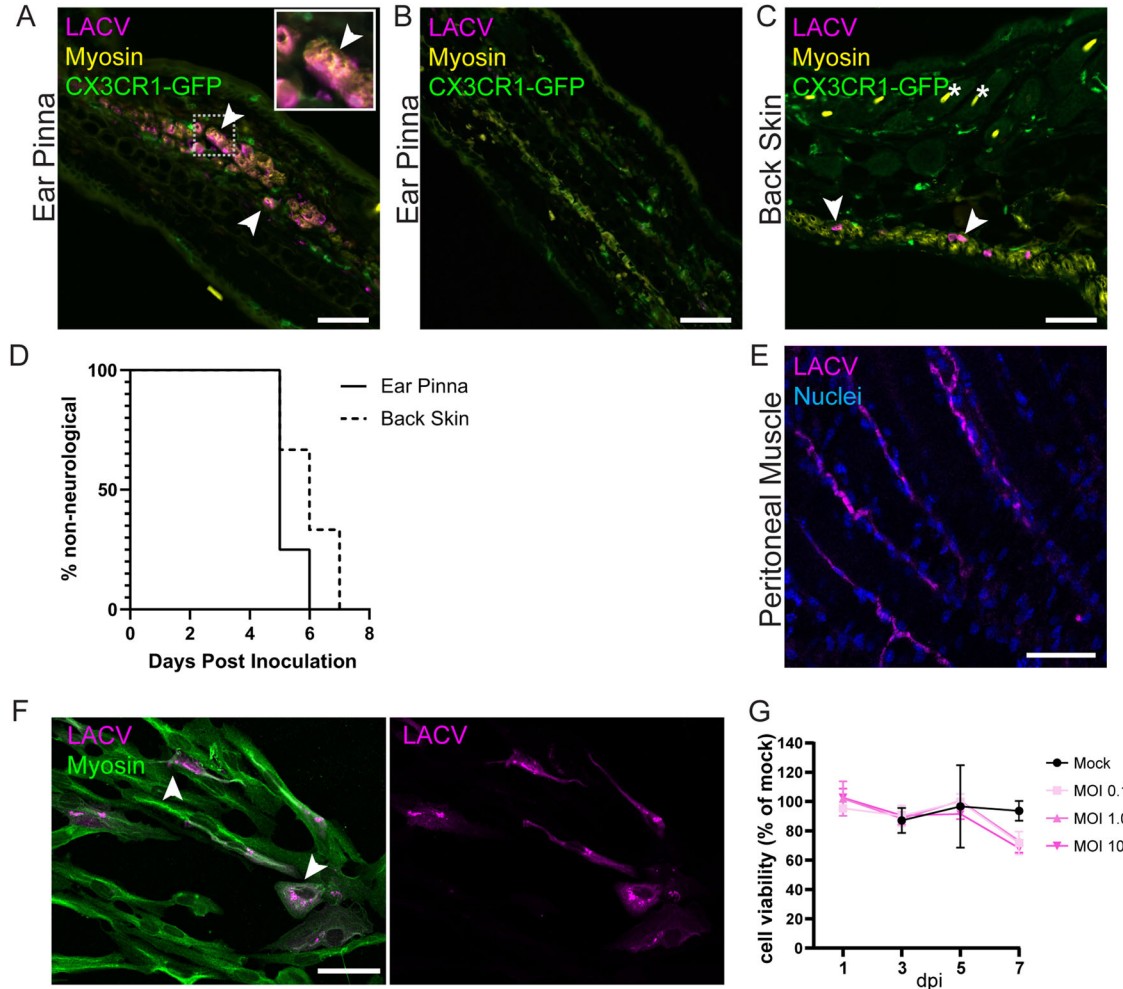

**Fig. 3 | Muscle infection is independent of virus injection location or species.** Mice were infected with $1 \times 10^5$ PFU LACV into the ear pinna (**A**, **B**, **D**), dermal back skin (**C**, **D**), or peritoneal cavity (**E**). Tissues, including muscle and skin, were removed at 48 hpi (**A**–**C**; $n = 3$ mice), 96 hpi (**E**; muscle only; $n = 3$ mice), or mice monitored for neurological disease (**D**). Fluorescence staining was performed with antibodies against muscle myosin (yellow, **A**–**C**; or green, **F**), LACV (magenta, **A**–**C**, **E**, **F**), GFP (green, **A**–**C**), or Hoechst dye for nuclei (blue, **E**) and representative images are shown from two independent staining experiments. Human muscle myoblasts were infected with LACV and infection evaluated by microscopy (**F**; multiplicity of infection (MOI) = 0.1) and cell viability monitored with PrestoBlue assay (**G**; $n = 2$ mock wells per time point and $n = 3$ LACV wells per time point). Data graphed as mean with error bars representing SD for $n =$ . Inset image for (**A**) is 2.5x zoom. Arrowheads indicate cells positive for both myosin and LACV. Asterisks mark autofluorescent hair follicles. For neurological disease development, $n = 3$–4 mice across 2 cohorts. **A**, **B**, **D** are representative of 3 mice from each injection method. Scale bars represent 50 µm. Source data are provided as a Source Data file.

intensity indicated by arrows) and around hair follicles in the skin (second row, asterisks), indicative of normal follicular growth. However, there was an absence of active caspase-3 staining within or around LACV-infected muscle cells in the skin (Fig. 4A second row arrows) or thigh muscle (third row). Interestingly, the lack of muscle cell death was observed in the PC regardless of whether there was immune cell infiltration (Fig. 4A, middle row merge, IC label), suggesting that immune cell recruitment to the site of infection is not responsible for inducing muscle cell apoptosis.

Despite the PC producing virus for multiple days post-injection, the low overall number of myofibers infected makes it difficult to extensively evaluate cell death in vivo. To evaluate if LACV is capable of inducing cell death of muscle cells, we removed the PC from the skin of naïve mice and infected it with LACV ex vivo to increase the incidence of virus infection. These skin muscle ex vivo (SMEV) cultures were productively infected with LACV, produced viable virus, and had a high rate of muscle cell infection (Fig. S4). SMEV cultures were infected with LACV for 48 h and analyzed for LACV and active caspase-3 staining. Despite robust infection of SMEV cells with LACV, only rare active caspase-3 cells were detected, with no

difference compared to mock-infected SMEVs (Fig. 4B). Furthermore, a cell viability assay conducted over 7 days showed no observable loss in cell viability for LACV-infected SMEVs compared to mock infected SMEVs (Fig. 4C). This is in direct contrast to LACV infection of neurons ex vivo[22] or infection of cerebral organoids[24], where substantial active caspase-3 staining and cell death is observed. Thus, the consequence of LACV infection of muscle cells appears to differ significantly from the LACV infection of neurons.

**Muscle cells delay the release of mature virus, which appear to be retained in the cis-Golgi**

Given the differing fates of neuronal lineage cells and muscle cells for survival during LACV infection, we hypothesized that aspects of virus production or replication might be altered as well. We first compared viral release kinetics by harvesting supernatant from LACV infected SMEV cultures or human-derived neural stem cells (NSCs) and performing plaque assays on viable virus. Within the first 24 hpi (1 dpi), NSCs released 5 log more virus than SMEV cultures and maintained this level of virus release to 48 hpi (2 dpi) even as NSCs are rapidly dying (Fig. 5A). In contrast, SMEV

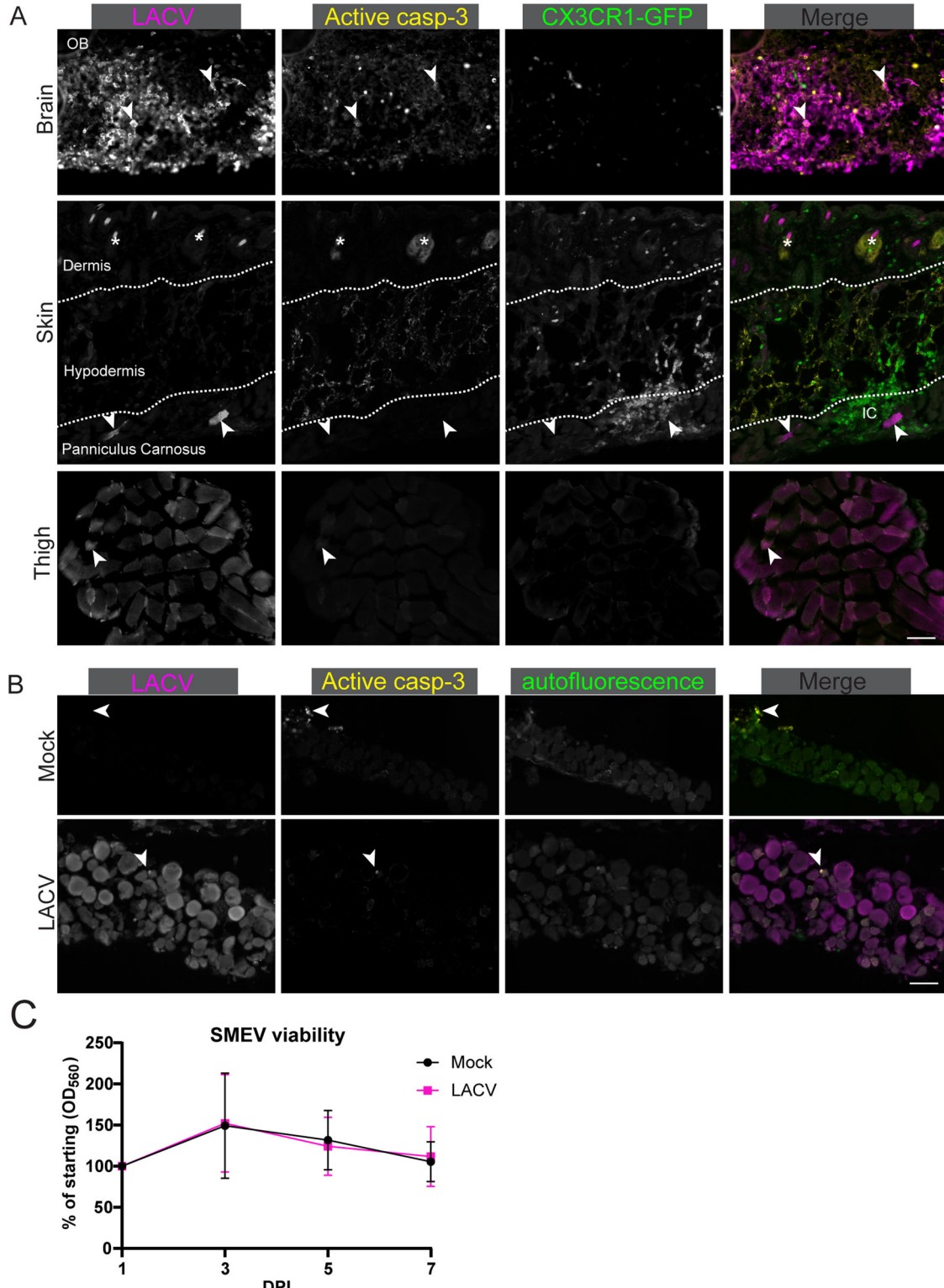

**Fig. 4 | LACV does not induce muscle cell death.** Mouse brain, skin, and ipsilateral quadricep (thigh) muscle (**A**) or ex vivo skin muscle (**B**) was stained with antibodies against LACV (magenta) or active caspase-3 (white). In (**A**) arrowheads indicate LACV infected cells, asterisks demarcate auto-fluorescent hair follicles, and dotted line shows boundaries between dermis, hypodermis, and panniculus carnosus. In (**B**) arrowheads indicate active caspase-3 positive staining. Scale bars represent 50 μm. OB Olfactory bulb. IC denotes region of immune cell accumulation (**A**, middle row). Viability of SMEVs was assessed by PrestoBlue assay to measure mitochondrial function every 2 days in mock and LACV infected tissues (**C**). Micrographs are representative of $n = 2$ (**A**, brain), 3 (**B**), or 4 (**A** skin, thigh) biological replicates. **C** Means are plotted with SD; $n = 10$ biological replicates per time point from two independent experiments. Source data are provided as a Source Data file.

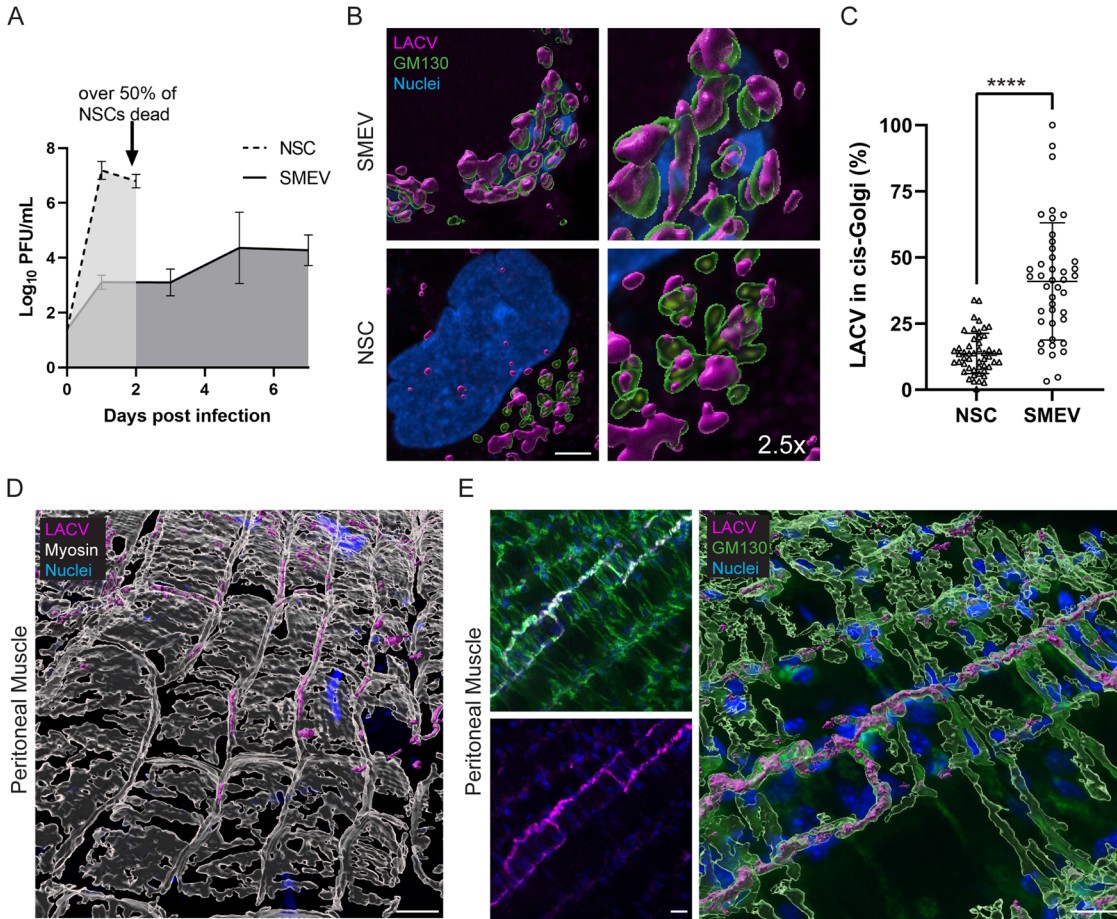

**Fig. 5 | LACV is retained in the cis-Golgi in muscle cells.** Neural stem cells (NSCs) and SMEVs were infected with LACV and released virus was quantified in the supernatant by plaque assay (**A**; *n* = 6 biological replicates per time point for each sample type). Golgi staining (GM130; green) was performed in infected SMEVs (5 dpi) and NSCs (1 dpi) (**B**) and LACV (magenta) occupying the cis-Golgi was quantified as a proportion of virus contained within the cis-Golgi compartment demarcated by GM130 compared to total LACV within individual cells (**C**). Peritoneal muscle from mice infected IP in vivo was removed and stained for LACV (**D**, **E**; magenta) and myosin (**D**; white) or GM130 (**E**; green). The 3D volumes of images from (**B**, **D**, and **E**) were rendered for visualization as partially transparent overlays (GM130, myosin) or fully opaque structures (LACV). For (**C**), 2 SMEV tissues (*n* = 42 cells) and 4 replicate NSC wells (*n* = 50 cells) were quantified. For (**D**, **E**), images are representative of peritoneal muscle from 2 mice. **A**, **C** Data plotted as means with error bars representing SD. ****$p < 0.0001$ ($p = 2.15 \times 10^{-12}$) by unpaired, two-tailed Student's *t* test. Scale bars represent (**B**) 5 μm, (**D**) 20 μm and (**E**) 10 μm. Source data are provided as a Source Data file.

cultures slowly released virus, failing to display the logarithmic release kinetics typically observed for highly virus permissive cells like neurons (Fig. 5A). LACV is processed through the Golgi apparatus before release. Ojha et al. demonstrated that retention of LACV within the Golgi body results in decreased virus release from neuronal cells and reduced cell death[25]. We therefore performed Golgi staining in SMEVs and NSCs to compare how much LACV was retained within the Golgi during infection. We quantified the proportion of virus contained within the cis-Golgi compartment demarcated by GM130, as a proportion of the total virus signal within individual cells (Fig. 5B, C). SMEVs had more LACV retained within the cis-Golgi compared to NSCs. This was true even comparing the cells at peak viral release at 1 dpi for NSCs compared to 5 dpi for SMEVs (Fig. 5B, C).

Finally, to confirm that our SMEVs were accurately modeling an in vivo infection, we verified the Golgi occupancy phenotype using our mouse model. We examined peritoneal muscle at 4 dpi by confocal microscopy. Virus was localized to the periplasmic space surrounding the actin-myosin bundles at the center of the myofibers (Fig. 5D). We then performed GM130 staining and found that LACV is largely localized to Golgi structures that decorate the periphery of the myofibers and cross across the top of the muscle cells (Fig. 5E).

## Discussion

Arboviral encephalitis can only occur by dissemination from the skin bite site to the central nervous system. For many neurotropic arboviruses, the cell type infected in the skin is often keratinocytes or skin-resident innate immune cells, which become activated and traffic the virus to the descending lymph node, where virus disseminates to the blood and the CNS. In the current study, we found a different system for orthobunyaviruses. LACV preferentially infects and replicates within muscle cells connected to the skin, including the PC and auricular muscle, with a lack of infection in other skin resident cells or motile immune cells. This suggests that the initial events in the skin following bunyavirus transmission are different from that seen with other arboviruses, and use a distinct mechanism of virus dissemination. Indeed, we were not able to locate any prior references to skin-associated muscle infection for any arbovirus or viral pathogen and we did not observe PC infection with WNV. It is notable however that limited studies have looked at skin muscle during arbovirus infection and it is not readily analyzed by flow cytometry or single cell analysis for large data analysis.

Other arboviruses have been shown to infect skeletal muscle, but primarily skeletal muscle of the limbs and smooth muscle of the heart[11,13,26,27]. Indeed, Chikungunya, Dengue, and Zika have all been

reported to infect skeletal muscle across animal models and human samples[13,26,28–31]. Approximately 25% of clinical human DENV and CHIKV patients have myocardial involvement, indicating that arboviral infections can readily occur in the heart[32]. We detected high levels of LACV transcripts in large muscle tissues in our mouse model including the thigh muscle and heart in later stages of infection. Interestingly, this infection was not associated with damage of muscle cells or inflammation in those areas. Indeed, LACV may not induce the muscle cell damage observed by some other arboviruses. CHIKV and DENV induce damage and necrosis of muscle tissue, which contributes to the muscle pain associated with these infections[13,26]. When CHIKV replication is restricted from muscle cells with an interfering miRNA, infiltration of immune cells decline, cytokine expression is reduced, and joint inflammation is suppressed—all without an increase in viral titer[26]. Thus, the lack of muscle cell death associated with LACV may also explain the lack of inflammation. Understanding why these viruses differ in damage to muscle cells may provide valuable information on the pathogenesis associated with CHIKV and DENV infection of the skin.

The lack of damage and apoptosis following LACV infection is also very different from what is observed with LACV infection in the brain, where virus infection induces widespread apoptosis of neurons and recruitment of large numbers of inflammatory cells. Our investigation into the differences between muscle cells and neurons suggest that virus trafficking through organelles may contribute to the difference in cell death. LACV release requires trafficking through the Golgi apparatus[25,33]. We found there is a retention of virus within the Golgi body of muscle cells in the PC as compared to neurons and this correlates to a diminished release of virus from muscle cells into the surrounding supernatant (Fig. 5). Although this was not a direct comparison of neurons within brain tissue, which were too difficult to separate out for this type of analysis, the differences in virus localization that were observed may provide intriguing explanation for the differences in apoptosis. Muscle cells may have slower virus release compared to highly permissive cells like neurons, which could be related to why LACV infection does not induce substantial death of muscle cells. With reduced demands on the host cell machinery to rapidly replicate and release virus, the cells may be less prone to initiate apoptosis. Other studies have shown that the Golgi apparatus is substantially different between muscles and neurons. For example, NSCs contain all three Golgi compartments, including cis, medial and trans Golgi. In contrast, only the cis-Golgi (stained by GM130) has been confirmed to be present in skeletal muscle cells, and the morphology of the compartment varies based on muscle fiber type[34–37]. Thus, basic cell organelle structure that differs between muscle cells and neurons may lead to different outcomes of LACV infection in these cell types, with muscle cells in the skin being a slowly replicating reservoir of virus.

In the case of LACV, PC cells may be a key factor in establishing early infection that can then lead to CNS invasion. The PC is largely understudied. It is a highly regenerative tissue, with an over-abundance of muscle stem cells compared to other skeletal muscle[38,39]. In rodents, the PC covers the majority of the midsection, with a notable lack of coverage in the extremities[20], suggesting foot-pad infection in mice may not mimic skin with associated thin bands of muscle. The PC in humans covers a relatively small area but is found in skin of the face, scalp, neck, and palms, with varied vestigial segments found on the torso[20]. Occasionally, the PC is also found in cadaver and patient samples in other locations including the heels of the feet and the abdominal skin, but this is considered atypical in the larger population[40–42]. Our data suggest the skin-associated muscle may serve as a replication niche following skin infection where layers like the PC are present. It is interesting to speculate if this also occurs in human infection, as the PC is coincidentally found in body locations most frequently left exposed to mosquitos like the face and palms and therefore best positioned to become infected following a bite.

## Methods

### Mouse infections

Animal experiments were conducted in accordance with animal protocols approved by the RML Animal Care and Use Committee. Male and female $Ccr2^{RFP/+}$ $Cx3cr1^{GFP/+}$ heterozygous mice were generated by crossing $Ccr2^{RFP/RFP}$ mice (Jax strain 017586) with $Cx3cr1^{GFP/GFP}$ mice (Jax strain 005582). HET mice and C57BL/6J mice were bred in-house and used after weaning at 20–23 days post birth. Mice in different litters were randomly assigned to different groups. Mice were housed with a 12 h light/dark cycle at a temperature range of 21–24 °C and a humidity range system set to 50% ± 10%, and were maintained on 2016 Teklad global rodent chow. Mice were provided food and water ad libitum. Infections were performed using human LACV 1978 stock (a gift from Richard Bennett; NIAID/NIH) propagated in Vero cells and described previously (Bennett et al.[2], Bennett et al.[43]) or West Nile virus (strain 385-99; BEI resources) propagated in Vero cells to match LACV stocks. For mouse infections, sterile phosphate buffered saline (PBS) containing $10^5$ PFU virus was injected intradermally into the shaved back skin above the left hind flank (25 µL volume) or into the ear pinna (10 µL volume) using a 29-gauge insulin syringe with the bevel facing up (flank) or a 32-gauge Hamilton syringe (ear). Consistent and correct intradermal injection was confirmed by retention of delivery volume in the skin. An equal dilution and amount of cell culture media was used for mock control mice for each injection location. Skin for analysis was removed from the original injection site as a full-thickness biopsy or whole ear unless otherwise noted.

### Mosquito salivary gland extract isolation

*Aedes aegypti* (Liverpool strain, LVP) mosquitoes were reared under standard insectary conditions (27 °C, 80% humidity, 12-h light/dark cycle) in the Laboratory of Malaria and Vector Research, NIAID, NIH. *Aedes triseriatus* mosquitoes were reared as previously described[44]. Salivary glands from sugar-fed adult female mosquitoes (5–7 days old) were dissected in PBS, pH 7.4 under a stereomicroscope. Salivary gland extract (SGE) was obtained by disrupting the gland walls by sonication (Branson Sonifier 450). Tubes were centrifuged at $12,000 \times g$ for 5 min, and supernatants were kept at −80 °C until used. In total, 1.7 µg of SGE was used to represent the amount of protein delivered per mosquito bite.

### Skin muscle ex vivo cultures

Mouse back skin was removed and pinned, hair side down, into a wax dissecting tray containing sterile PBS. Skin-associated muscle tissue was excised as intact muscle sheets from the dermis using a scalpel and tweezers and transferred to a 24-well plate as ~1 cm² pieces per well. Virus or mock culture media were applied to muscle cultures equaling $2 \times 10^5$ PFU (PrestoBlue assay) or $4 \times 10^6$ PFU (microscopy) per muscle segment in DMEM (GIBCO) containing 10% FBS (Atlas Biologicals). Culture media containing virus was removed after 2 h, muscle section rinsed with fresh media, and replaced with fresh culture media (20% FBS in DMEM) for the remainder of the culturing period.

### SMEV viability assay

PrestoBlue assay was performed on SMEVs to assess mitochondrial respiration. Briefly, supernatant was removed from SMEV cultures and replaced with 1x PrestoBlue reagent (Invitrogen) diluted in muscle culture media. After 30 min incubation at 37 °C with gentle rocking, PrestoBlue reagent was removed from the wells and plated in triplicate in a flat-bottomed 96-well plate. The plate was read for fluorescence using 560 nm excitation and 590 nm emission in an optical plate reader (BioTek) with Gen5 software. Blank wells of PrestoBlue reagent incubated without exposure to muscle sheets were used to subtract

background reagent fluorescence. For each sample, the PrestoBlue result at each timepoint was normalized to the starting (1 dpi) value to yield the percent change over time.

## Plaque assays

Skin and muscle tissues were homogenized with ceramic beads in a bead beater machine (BeadMill 24; Fisher Scientific) and clarified by centrifugation. Skin was homogenized at 6000 rpm for 1 min. All other tissues were homogenized at 5200 rpm in two cycles of 20 s each with a 5 s dwell time between cycles. Tissue homogenate, plasma, and cell culture supernatants were diluted in 2% DMEM and plated on confluent Vero cells for 1 h at 37 °C. Wells were overlaid with 1.5% carboxyl methylcellulose (Sigma) in Minimal Essential Medium (MEM; Gibco) until plaques reached sufficient size for counting. Wells were fixed with 10% formaldehyde in 1x PBS for 1 h and cell monolayers stained with 35% crystal violet dye. Plaque numbers were normalized to original tissue weights or per mL for plasma and cell culture supernatant. Data were $log_{10}$ transformed and graphed in Prism v. 9.3.1 (GraphPad) as mean ± standard deviation with each point representing each mouse.

## qPCR for La Crosse RNA

Tissues were homogenized in TRIzol reagent (Life Technologies) using ceramic beads in a bead beater instrument as described for plaque assays. Isolation of RNA was performed per the TRIzol reagent protocol as described previously followed by genomic DNase digestion (Invitrogen) and column cleanup (Zymo). cDNA was synthesized using the iScript cDNA Synthesis kit (BioRad). qPCR was performed using the SYBR Green kit (BD Life Sciences) and primers against the s segment of LACV [forward 5′-ATTCTACCCGCTGACCATTG-3′, reverse 5′-GTGAGAGTGCCCATAGCGTTG-3′] and mouse GAPDH [forward 5′-AACGACCCCTTCATTGAC-3′, reverse 5′-TCCACGACATACTCAGCA-3′]. To ensure primer specificity, a BLAST search (NCBI) was performed on each primer set and primers were tested on known positive samples and no RT controls to confirm amplification of a single product as expected without mis-amplification. Plotted data were calculated as the percent difference in threshold cycle (CT) value ($\Delta CT = CT$ for GAPDH gene − CT for specified gene) and normalized as the % of GAPDH. Data were graphed in Prism v. 8.2.0 (GraphPad) as mean ± standard deviation (SD) with each point representing each mouse.

## Neural stem cell culturing

Human H9 neural stem cells (Thermo Fisher Scientific; N7800) were maintained for less than 12 passages in KnockOut™ D-MEM/F-12 media containing StemPro® Neural Supplement, GlutaMAX-I Supplement (Thermo Fisher Scientific), Fibroblast growth factor and Epidermal growth factor. For PFU analysis, neurospheres were infected with LACV at an MOI of 0.1. Supernatant was isolated from neurospheres every 24 h by centrifugation and assayed for infectious virus by plaque assay.

For microscopy, a coverslip-bottom 8-well chamber coverglass (Nunc) was coated overnight with 40 μg/mL fibronectin in PBS. Neural stem cells were plated as single cells and allowed to reach ~80% confluence before infection with LACV at an MOI of 0.1. Cells were washed gently and fixed with 4% (w/v) formaldehyde for 30 min and stored in PBS before staining for microscopy as described below.

## Generation of human myoblast immortalized cells

Immortalized myoblast (M007) cells were derived from primary muscle biopsies under a protocol approved by the University of Minnesota Institutional Review Board (IRB). Primary cultures were established in F10 medium (HyClone) supplemented with 20% FBS (Peak Serum), β-mercaptoethanol 1x (Gibco), $10^{-9}$ M dexamethasone (Sigma), 10 ng/mL bFGF (Peprotech), Glutamax 1x (Gibco), and Penicillin/Streptomycin 1x (Gibco) with culture at 37 °C and 5% $O_2$/5% $CO_2$. Cells were then sorted into CD56+ (myoblast) and CD56− (fibroblast) fractions on a BD FACSAria (BD Biosciences) using APC-

conjugated CD56 monoclonal (17-0567-42, eBioscience). Myoblast cultures were transduced with immortalization lentivectors pbabe-cyclinD1 + CDK4R24C (gift of Christopher Counter, Addgene)[45] and pLV-hTERT-IRES-hygro (gift of Tobias Meyer, Addgene)[46]. Cells were cultured past senescence of control untransduced cultures and then subcloned by single cell sorting into 96-well plates using the FAC-SAria. Individual clones were expanded and karyotyped at the University of Minnesota Cytogenomics Core. M007.2 displayed 46,XX (normal human female) karyotypes. Myoblasts maintained in the same media used to culture primary cells. LACV infections were performed as described for NSC cells using an MOI of 0.1 (microscopy) and a range of MOIs (0.1, 1, 10) for viability assay with PrestoBlue.

## Microscopy

Tissues were fixed overnight (injection site biopsy skin and peritoneal muscle) or for 1 h (SMEV cultures) in 10% (v/v) formalin. Peritoneal muscle was washed with 1x PBS and stained with antibodies immediately following fixation. Other tissues were cryopreserved in 30% (w/v) sucrose in PBS until tissues sank to the bottom of the tube. Skin and SMEVs were embedded in OCT freezing medium (Sakura Finetek USA) and frozen on dry ice. In total, 10–12 μm thick sections were taken by cryostat and mounted on slides.

For fluorescence imaging, mounted sections and full-thickness peritoneal muscle segments were blocked with blocking buffer (PBS with 1% BSA, 0.3 M glycine, 0.1% fish skin gelatin, 0.1% Triton X-100 and 2% donkey serum) for 30 min and immunostaining was performed with overnight incubation of primary antibodies diluted in blocking buffer at room temperature (see Table S1). After PBS washes, secondary antibodies diluted in blocking buffer were left on tissue sections for 2–4 h at room temperature (see Table S1). Hoechst 33342 (Tocris Bioscience) was used at 1 μg/mL to stain DNA in the first of 3 washes in PBS, where noted in figures. Coverslips were mounted with Prolong Gold (Thermo Fisher; skin mounts and SMEV phenotyping) or Prolong Diamond (Thermo Fisher; GM130 experiments) and allowed to cure before imaging. One additional slide was stained with only secondary antibodies in each batch of staining to confirm any positive signal was not the result of non-specific binding from secondary antibodies. All imaging was performed at room temperature. Large tile scans of skin and heart were acquired using a wide-field epifluorescence slide scanning microscope (Axio Scan.Z1; Carl Zeiss Microscopy) driven by ZEN Blue v3.1 with a Plan-Apochromat 20X/0.8 objective. Golgi imaging was performed on a point scanning confocal microscope (LSM 880; Carl Zeiss Microscopy) driven by ZEN Black v.2.3 software with an EC Plan Neofluar 40X/1.3 Oil DIC objective, and immersion oil at a refractive index of 1.518. Stacks were collected with a lateral resolution of 0.1 μm and z-spacing of 0.25–0.30 μm. Peritoneal muscle was mounted to the bottom of an imaging dish (MatTek) using 4% agarose for confocal microscopy with a Plan Apochromat 20X/0.8 objective, a lateral resolution of 0.21 μm and z-spacing of 0.73–0.85 μm.

In situ hybridization was performed on sectioned skin using the RNAScope 2.5 HD Assay-RED kit (Advanced Cell Diagnostics Inc) on 12 μm frozen sections without antigen retrieval, per manufacturer's instructions. ACD custom designed probes were used to distinguish the genomic and replicative strand of La Crosse virus. Hematoxylin staining was used to visualize nuclei (Vector Labs) and images acquired using the Axio Scan.Z1 as described above.

Optically cleared skin from the injection site was generated using the iDISCO protocol[47] (https://idisco.info/idisco-protocol/) with the following changes Whole skin biopsies were treated as described for adult organs, with a base incubation time ($n$) of 4 days for incubation steps. At final clearing steps post dehydration, 66% DMC/33% methanol incubation was shortened to 15 min before 100% DCM incubation. Prior to clearing, skin was stained with primary antibodies [chicken

anti-GFP, rabbit anti-LACV, and mouse anti-myosin] (noted in Table S1). Anti-chicken AF 488, anti-rabbit AF 647, and anti-mouse AF 594 were used as secondary antibodies (Table S1). Cleared skin was imaged in an imaging dish (MatTek) filled with dibenzyl ether (Sigma). Tissue was imaged at room temperature on a point scanning confocal microscope (LSM 880; Carl Zeiss Microscopy) driven by ZEN Black v.2.3 software with a Plan Apochromat 20X/0.8 objective, with a lateral resolution of 0.28 μm and z-spacing of 2 μm.

### LACV infection quantification
LACV infection quantification was performed using FIJI/ImageJ[48] (version 1.53t). A threshold was applied to Single channel images to define positive signal (250-65535 for LACV; 200-65535 for myosin). Quantification of the proportion of skin muscle cells infected with LACV was performed by manually counting individual cells located below the hypodermis that were LACV positive and dividing by the total number of cells that were myosin positive. For the heart, a box was drawn as closely as possible around the boundaries of the tissue in the stitched tile scans and the myosin mean signal intensity and LACV mean signal intensity were quantified using the measure function. A ratio was calculated of myosin mean intensity divided by LACV mean intensity for each tissue section and plotted as an individual point in GraphPad Prism (9.3.1). This was repeated for 3–4 heart sections from two mice. The same acquisition settings were used for all heart sections and images were collected in a single imaging session.

### Image deconvolution and LACV Golgi occupancy quantification
Deconvolution was performed on image stacks with Huygens Professional (Scientific Volume Imaging, v 22.10) using the CMLE algorithm with a maximum of 40 iterations and signal-to-noise ratio of 20. Deconvolved image stacks were imported into Imaris x86_64 v.10.0.0 (Bitplane AG) for analysis and rendering. The GM130 channel was segmented and rendered using the Surface module (grain size = 0.088 μm, threshold 350–2000). The Spots module was used to calculate the total LACV intensity within each infected cell. To calculate the integrated intensity of LACV signal within the Golgi, a distance transformation filter was applied, and the LACV spots were filtered with an upper threshold of 0.05 μm as the shortest distance to the rendered GM130 surface. The percentage of LACV within the Golgi was calculated as the sum intensity of LACV within the GM130 (Golgi) rendered volume divided by the LACV sum intensity across the entire cell, on a per cell basis. At each step, the accuracy of surface or spot thresholding was manually verified for each cell analyzed ($n = 50$ cells for NSCs, $n = 42$ cells for SMEVs, with 5–6 fields of view across 2 replicate experiments). Data were graphed in Prism v. 9.3.1 (GraphPad) as mean ± standard deviation with each point representing a single cell. Groups were compared by unpaired t-test, with statistical significance indicated with asterisks.

### Peritoneal muscle surface rendering
Image stacks were imported into Imaris x86_64 v.10.0.0 (Bitplane AG) for rendering volume reconstructions for visualization and analysis. The GM130, myosin, and LACV channels were segmented and rendered using the Surface module (grain size = 0.088 μm, thresholds of 200–2000 [GM130], 550–1560 [myosin], and 350–1500 [LACV]) to match the fluorescence signals.

### Reporting summary
Further information on research design is available in the Nature Portfolio Reporting Summary linked to this article.

## Data availability
Source data are provided with this paper.

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

## Acknowledgements

The authors thank Sue Priola and Audrey Chong for their critical reading of the manuscript as well as the entire Peterson Lab for feedback and suggestions. We thank Ryan Kissinger (NIH medical illustrator) for creating the mouse illustration in Fig. 2. We also thank the Rocky Mountain Veterinary Branch, especially Jeff Severson, Shelby Heinz and Sandra Gleason, for their excellent animal husbandry, breeding, and assistance. This work was supported by the Division of Intramural Research, National Institute of Allergy and Infectious Disease through AI001102-11 (K.E.P.). There was no input from funders in the study design, data collection and analysis, decision to publish, or preparation of the manuscript.

## Author contributions

C.A.S., E.C. and K.E.P. designed and developed the project, C.A.S. and J.M.L. designed and conducted the experiments and analyzed the data, P.C.V. isolated the salivary gland extracts for inoculations, N.A.G., E.A.T., D.B. and M.K. generated and characterized the muscle cell line M007, C.A.S. wrote the original draft of manuscript, all authors contributed to editing the manuscript.

## Competing interests

The authors declare no competing interests.
