## [Peer Review File · Nature Communications]

Skin muscle is the initial site of viral replication for arboviral bunyavirus infection.REVIEWER COMMENTS

Reviewer #1 (Remarks to the Author):

In this study, the authors set out to characterize which cells are infected in the skin by La Crosse virus (LACV), a critical question in the field. They use an intradermal (ID) inoculation route in weanling mice and show by immunofluorescence the presence of LACV antigen and RNA in cells of the skin, which based on morphology and myosin staining they conclude are skin muscle cells. They then go on to show that muscle infection is independent of inoculation route and that ID infection leads to a disseminating infection to other organs include the heart and peritoneal muscle where they find viral RNA by qPCR. To begin to understand how LACV may influence skin muscle cells, they develop a very cool ex vivo model of skin muscle. They show that LACV infects this model, yet does not lead to caspase-3 dependent cell death. Finally by comparing mouse muscle explants to human neuronal stem cells they show that LACV is retained in the cis-golgi.

I think this a great start to a major gap in our knowledge. There are several strengths to this manuscript including the overall questions, the use of ID inoculation (which is rarely used for LACV) and establishing an ex vivo model to study LACV muscle infection. However, there are several concerns outlined below that should be addressed for publication, including a major concerns regarding controls and the conclusion they draw from their experiments.

1. A major conclusion of this manuscript is that LACV infects skin muscle cells which is dramatically different than other arboviruses (CHIKV, WNV etc). This is an important observation, and if you want to make this conclusion other virus controls in the same mouse system are needed. I suggest adding another arbovirus that infects other cells to show this difference.

2. The authors conclude based on myosin staining that LACV infects muscle cells. You're probably right but more controls are necessary to define this tropism. I suggest including staining for other skin cells in this model to show LACV is NOT in these cells. Important markers are going to be needed for skin fibroblasts, langerhan cells, and keratinocytes. You will also need to stain for other cells in the heart. From a quick search it looks like Myosin I is expressed in some fibroblasts (PMID: 8449984) so these other cell markers are important to map this out.

3. While these studies in mice are nice, it leaves us wondering if the same cells get infected in human skin. There are many fibroblast and muscle cell lines available (mouse and human), as well as available human skin models that have been used for work with other arboviruses (PMID: 10888933). These cell lines or human models in combination with other arboviruses would be great to nail your point home and significantly strengthen your story. I suggest adding them.

4. In Figure 2, you don't find RNA in the skin at time = 0? Or does N.D. mean not determined? I couldn't find this in the figure legend. If not determined, determine please. It is also interesting you are finding RNA and infectious particles in the ipsilateral thigh 10 mins post inoculation. Can you explain why you think this is?

5. The premise of Figure 4 is to address 1) if LACV kills muscle cells and 2) if the cell death mechanism in skin muscle cells is similar to neurons. If you want to make these comparisons, you will need to include neuronal cultures or brain slices from the same mice to show 1) LACV kills neurons but not muscle cells and 2) caspase-3 is being induced in these mice in a differential way. I think this is also an important technical control. From the images in Figure 4 there is no caspase-3 staining so it would be nice to show us an image of what massive staining looks like. Also, in reference 18, you use a colorimetric assay for caspase-3 expression, is that more sensitive than antibody staining? Finally, given that reference 18 used IP infection of LACV and this study is establishing a new ID model, it is important to show that LACV does induce Caspase-3 dependent death in neurons after ID inoculation.

6. Figure 5 is really cool! However, using two different culture models from different hosts makes the results hard to interpret. While you may be right, it will be important to generate neuronal cultures from the same mice to compare with your SMEV cultures. Cell lines could also be used here.

Reviewer #2 (Remarks to the Author):

In this manuscript by Schneider et al., the authors investigate the first step in the pathogenesis of La Cross virus in a mouse model system. They found that the virus infects and replicates nearly exclusively within skin-associated muscle cells. The manuscript addresses a critical yet highly understudied area of the initial transmission of the virus: what are the first target cells of infection, and how does the virus spread from the initial site to other parts of the body?

The experimental design is clear, and the interpretation of the results is sound. Furthermore, I want to compliment the authors' excellent histopathology and immunohistochemistry work. I believe that the data presented here is exciting and offers a novel view of the initial replication site of LACV. However, there are specific gaps/weaknesses in the conclusion that the authors draw for the pathogenesis from the findings. Expanding the discussion can quickly address some of those, whereas others need to be handled with additional experiments.

I agree with the authors that an intradermal inoculation is more representative of a mosquito bite than a subcutaneous inoculation. Nevertheless, a vast body of literature shows mosquito saliva's impact on virus replication and dissemination for many viruses. This is not discussed in this manuscript at all and must be added. Mosquito saliva could, for example, attract other cells through chemotaxis, such as macrophages, dendritic cells, and other cell types, within hours. Mosquito saliva can also influence the immunological state of other cells in the bite area. Why were no additional studies conducted with infected mosquitoes? Or at least mosquito saliva collected and mixed with the virus inoculum?

I also disagree with the authors that 100,000 PFU of challenge dose in 25 μ L mimics mosquito transmission. Although the tighter injection by the mosquito is unknown, I highly doubt it is 10e5 PFU is injected. Plus, mosquitoes inject just a few microliters close to a blood capillary in the dermis and definitely do not create the hydrostatic pressure of 25 microliters between the dermis and subcutis.

The authors state that no virus was detected at the injection sites early on, yet it was discovered in close-by muscle structures and RNA was detected in lymph nodes. How much of the initial inoculum is washed out hematogenously and infects other cells once in the bloodstream or the lymphatics? I do not question the findings here that there's long-term replication in the muscle structures. Still, I'm just wondering how much other cells in other tissues play a role in maintaining viremia that is overlooked by this experiment. It is necessary to do a titration study and use lower doses of the virus to see if similar results can be found.

-Has the natural history of LACV in the Cx3cr1GFP/Ccr2RFP mouse model been characterized? If yes, please add a reference to line 85 on page 3.

-In lines 276-278, the authors speculate that the PC also plays a role in human infections. This is a very interesting thought. However, the actual distribution of the PC muscle in humans is much smaller than the authors made it sound. This brings up the question if most of the mosquito bites in humans are not in areas where the PC is found, how do the data generated here play a role in human infection?

Technique Concern

Having done intradermal inoculations on many animal species myself, I know how difficult it is to deliver a precise amount of virus. Comparative anatomy tells us that the skin of mice is thinner than other mammalian species; plus, 3-week-old weanlings (very small) were used here. For example, was there additional liquid spilling out after the inoculation? Was a needle stepper used to deliver the accurate 25 microliters of volume, and if just a regular needle was used, how was it accounted for dead volume? What gauge needle was used with what bevel? More details need to be added in the material and methods section.

Reviewer #3 (Remarks to the Author):

The manuscript presented by Schneider et al. suggests, from an experimental mouse model, that skin muscle cells are the major site of replication of La Crosse virus, an emerging orthobunyavirus, and that the virus is capable of low-level replication over long periods in muscle cells, including distal ones.

Overall, this work concerns a subject of great interest, often neglected, namely the role of muscle cells in arbovirus infections.

This study is mainly based on a mouse model only. Whereas the authors state in the introduction that La Crosse virus is of high importance in human paediatric arboviral infections, it is regrettable that no study was conducted on human primary muscle cultures, given the known differences in susceptibility to certain arboviral infections between humans and mice. In this respect, no data is available on whether in vivo or in vitro, La Crosse virus, which seems to infect differentiated muscle fibres, is able to infect myoblasts/satellite cells.

The light/immunofluorescence microscopy images are of good quality. Unfortunately, the data produced concern very small microscopic fields, and at no time is there any mention of quantification or counting according to the usual methods in histology and anatomic pathology. Only pictures with few LACV immunoreactive cells are provided, without any quantification. Pictures from mock-infected animals are lacking in some figures. Although these data provide indications on interesting new findings, they do not provide sufficient information to conclude.

In addition, some errors, imprecisions presented in the different figures could be corrected and improved:

In fig.1, there is confusion in the Legend: whereas in Text (Results), line 99, and Legend Fig 1 line 6, Fig 1C concerns immunofluorescence, line 9 of the legend Fig 1 suggests that it corresponds to in situ hybridization...

In Fig 1D, how can the authors conclude that gRNA and rRNA are detected exclusively in skin-associated muscle, since no myosin labeling is mentioned?

Fig S1: it is not mentioned in the legend of the figure to which correspond the Figures S1 B' and S1B ""??

How can the authors say that "all images are representative of 2-3 mice across multiple sections"?? Does it concern 2 or 3 mice? What do the authors mean by "multiple sections? 10, 100, 1000? How can a single immunoreactive cell in a small field be representative?

Fig 4B: why are there magenta labeled cells in the "anti caspase 3" column in mock?

Figure 5 showing the retention of LACV in the Golgi apparatus would have benefited from a transmission electron microscopy approach

Figure S2: did the authors make the same experiments (testing susceptibility of isolated immune cells from the same mice towards LACV infection?)

Thus in general, the work does not support the conclusions and claims, and could prohibit publication.

REVIEWER COMMENTS

Reviewer #1 (Remarks to the Author):

We appreciate the kind comments by reviewer 1 and have addressed the individual concerns as stated below.

1. A major conclusion of this manuscript is that LACV infects skin muscle cells which is dramatically different than other arboviruses (CHIKV, WNV etc). This is an important observation, and if you want to make this conclusion other virus controls in the same mouse system are needed. I suggest adding another arbovirus that infects other cells to show this difference.

The reviewer makes an excellent point. We did a comparison with WNV (Supp. Fig. 2) and showed WNV infection was dramatically different. WNV did not infect the muscle layer in the skin and was primarily found above the panniculus carnosus in cells with morphology consistent with fibroblasts and epithelial cells of the hypodermis.

2. The authors conclude based on myosin staining that LACV infects muscle cells. You're probably right but more controls are necessary to define this tropism. I suggest including staining for other skin cells in this model to show LACV is NOT in these cells. Important markers are going to be needed for skin fibroblasts, langerhan cells, and keratinocytes. You will also need to stain for other cells in the heart. From a quick search it looks like Myosin I is expressed in some fibroblasts (PMID: 8449984) so these other cell markers are important to map this out.

Keratinocytes, and myeloid cells, including langerhans cells within the skin, are CX3CR1⁺ and thus are GFP positive in the mice that were used for the histological studies (Fig. 1A). We did not see any virus colocalization with cells stained for GFP in any of the imaging (Fig. 1A). In an additional study, using non-fluorescent C57BL/6 mice inoculated with LACV and LACV + *aedes triseriatus* salivary gland extract (SGE), skin tissue was stained with CD90, a mesenchymal cell marker (Fig. 1B). CD90 is expressed on fibroblast, myofibroblasts, and connective tissue, as well as keratinocytes, T cells, and stem cells. Importantly, CD90 is not expressed in mature myofibers. We did not see any colocalization with CD90 and LACV (Fig. 1B). Indeed, LACV was found only in myosin+ cells of the panniculus carnosus layer, demonstrating that virus infected myocytes.

We have also included better images of the heart, showing what are clearly muscle cells and have provided a no-primary control to demonstrate that positivity is not due to non-specific binding on antibodies (Supp. Fig. 3)

3. While these studies in mice are nice, it leaves us wondering if the same cells get infected in human skin. There are many fibroblast and muscle cell lines available (mouse and human), as well as available human skin models that have been used for work with other arboviruses (PMID: 10888933). These cell lines or human models in combination with other arboviruses would be great to nail your point home and significantly strengthen your story. I suggest adding them.

We appreciate the comment and were able to collaborate to attain human muscle cells. Although we observed strong infection in human muscle cells, there was only limited cell death at very late time points in culture (Fig. 3F, G).

4. In Figure 2, you don't find RNA in the skin at time = 0? Or does N.D. mean not determined? I couldn't find this in the figure legend. If not determined, determine please. It is also interesting you are finding RNA and infectious particles in the ipsilateral thigh 10 mins post inoculation. Can you explain why you think this is?

N.D. means not determined (which is in the revised manuscript). We did not perform the quantification by qPCR because the entire skin injection site was used in A for plaque assay. If we divide the skin injection site at the 0 hpi time point, the inoculum leaks out, resulting in very skewed results. However, the same mice were used in A and D, which shows replicative virus in that tissue at the 0 timepoint.

The rapid ipsilateral infection we believe to be the result of a small amount of inoculum seeping through the skin into the underlying muscle (ipsilateral thigh here). Based on the plaque assays (Fig 2A and 2B), this would be approximately 100-fold lower than what is observed in the skin inoculation site.

5. The premise of Figure 4 is to address 1) if LACV kills muscle cells and 2) if the cell death mechanism in skin muscle cells is similar to neurons. If you want to make these comparisons, you will need to include neuronal cultures or brain slices from the same mice to show 1) LACV kills neurons but not muscle cells and 2) caspase-3 is being induced in these mice in a differential way. I think this is also an important technical control. From the images in Figure 4 there is no caspase-3 staining so it would be nice to show us an image of what massive staining looks like. Also, in reference 18, you use a colorimetric assay for caspase-3 expression, is that more sensitive than antibody staining? Finally, given that reference 18 used IP infection of LACV and this study is establishing a new ID model, it is important to show that LACV does induce Caspase-3 dependent death in neurons after ID inoculation.

We have now included brain tissue from mice at 6 dpi, which show virus infection and caspase 3 staining of neurons (Fig. 4A) including strongly Caspase 3 positive (yellow co-stain) and weakly positive (arrows). In addition, Caspase-3 staining is observed in the hair follicles in the skin where continual cell turnover occurs (Fig. 4, second panel asterisks – the hair itself is auto fluorescent). In addition, we measured cell survival via presto-blue assay for human muscle cells (Fig. 3G), which shows only limited cell death in cultured muscles and not until seven days post infection.

6. Figure 5 is really cool! However, using two different culture models from different hosts makes the results hard to interpret. While you may be right, it will be important to generate neuronal cultures from the same mice to compare with your SMEV cultures. Cell lines could also be used here.

We did try to analyze Golgi staining in neurons from mouse brains as well as cerebral organoids. However, the complicated structure of neurons interspersed with other cell types made this extraordinarily difficult. We also looked at the Golgi in the human muscle myocytes, however, as these cells were not differentiated – they did not have the same highly distributed Golgi patterning as the muscle cells in tissue, which did not allow us to mimic what was observed *in vivo*. We did change the wording in our results and in the discussion to emphasize that other factors including species and model systems may be impacting these results.

Reviewer #2 (Remarks to the Author):

I agree with the authors that an intradermal inoculation is more representative of a mosquito bite than a subcutaneous inoculation. Nevertheless, a vast body of literature shows mosquito saliva's impact on virus replication and dissemination for many viruses. This is not discussed in this manuscript at all and must be added. Mosquito saliva could, for example, attract other cells through chemotaxis, such as macrophages, dendritic cells, and other cell types, within hours. Mosquito saliva can also influence the immunological state of other cells in the bite area. Why were no additional studies conducted with infected mosquitoes?

This would have been the best experimental design, but unfortunately, institutional regulations and availability of appropriate facilities limited our ability to use infected mosquitoes to infect animals.

Or at least mosquito saliva collected and mixed with the virus inoculum?

We agree with the reviewer and mixed virus with salivary gland extract from the appropriate mosquito species for LACV (*Aedes triseriatus*) (Fig. 1B) and WNV (*Aedes aegypti*) (Supp Fig. 2). Although there may be an increase in detection with the addition of saliva, in neither case did it affect which cells were infected.

I also disagree with the authors that 100,000 PFU of challenge dose in 25 μ L mimics mosquito transmission. Although the tighter injection by the mosquito is unknown, I highly doubt it is 10^5 PFU is injected. Plus, mosquitoes inject just a few microliters close to a blood capillary in the dermis and definitely do not create the hydrostatic pressure of 25 microliters between the dermis and subcutis.

It is true that injection of 25 μ L is greater than the volume delivered by mosquitoes, but it does allow us to ensure a correct ID inoculation was performed because we have a visible injection bolus that remains in the skin immediately after injection. We did see similar muscle infection with a 10 μ L volume into the ear pinna (Fig. 3). As to the infectious dose chosen for our experiments, we can't prove that 10^5 PFU would be delivered by a single mosquito bite, but this dose was also necessary for identifying the cell tropism by imaging. Lower doses would make it extremely difficult to achieve sufficient individual cell quantification without sectioning substantially into the skin of each animal.

The authors state that no virus was detected at the injection sites early on, but it was discovered in close-by muscle structures and RNA was detected in lymph nodes.

For Fig. 2D, N.D. means not determined (which is in the revised manuscript). We did not perform the quantification by qPCR because the entire skin injection site was used in Fig. 2A for plaque assay. If we divide the skin injection site at the 0 hpi time point, the inoculum leaks out, resulting in very skewed results. However, the same mice were used in A and D, where we show the initial infectious virus found at the injection site immediately after injection. We have clarified this in the revised manuscript.

How much of the initial inoculum is washed out hematogenously and infects other cells once in the bloodstream or the lymphatics? I do not question the findings here that there's long-term replication in the muscle structures. Still, I'm just wondering how much other cells in other tissues play a role in maintaining viremia that is overlooked by this experiment.

This is an interesting question. Previous work in our lab failed to detect viral replication within circulating blood cells and we could not transfer virus from infected to naïve animals through PBMCs or spleen cells (Winkler et al. *Acta Neuropath* 2015). This suggests that virus is not spread via immune cells, however, we cannot rule out that there may be another population of virus infected cells that we have not detected. We also cannot rule out that there is a small level of transmission to other muscle sites via vasculature or lymphatics to other sites following inoculation. Indeed, the detection of virus in the ipsilateral thigh at 0 hpi suggests that some virus gets out of the skin at inoculation, although this levels is 100 fold lower than that observed in the skin (Fig. 2 A vs B).

It is necessary to do a titration study and use lower doses of the virus to see if similar results can be found.

We did analyze 10^2 and 10^3 PFU for intradermal inoculation. Unfortunately, but not surprising given the sporadic nature of the muscle cell infection we see in the skin with 10^5 PFU, we did not locate virus-infected cells by immunohistochemistry at the lower doses. We did observe similar virus kinetics in tissues as with the 10^5 dose, but this was highly variable. The results with the lower doses were conducted by another trainee, who completed this analysis as part of a study on the influence of salivary gland extract on LACV pathogenesis, which is part of a separate manuscript that is currently in preparation.

-Has the natural history of LACV in the Cx3cr1GFP/Ccr2RFP mouse model been characterized? If yes, please add a reference to line 85 on page 3.

We have not published directly on the natural history of LACV in the Cx3cr1GFP/Ccr2RFP mice. We chose these mice so that we could potentially differentiate between dendritic cells and monocytes in the skin, if these cells were infected. However, studies with B6 mice and Cx3cr1GFP/Ccr2RFP mice used for IP or ID inoculations, we see no difference in the kinetics, incidence or pathology of LACV infection.

-In lines 276-278, the authors speculate that the PC also plays a role in human infections. This is a very interesting thought. However, the actual distribution of the PC muscle in humans is much smaller than the authors made it sound.

We are not sure what the reviewer is referring to in regard to our statements. The PC is located in humans in the regions we indicated, as shown in the cited papers for those statements.

this brings up the question if most of the mosquito bites in humans are not in areas where the PC is found, how do the data generated here play a role in human infection?

The PC is found in regions that would often be exposed to mosquito bites, including hands, neck and face, which are skin sites often exposed to mosquitos. Although it is interesting to speculate that bites in those regions may lead to more productive LACV infections, those may not be the only regions where there is access to muscle via the skin. In the mouse model, we also see infection with other muscle tissue including ear muscle and intraperitoneal muscle. This idea is also complicated by the low rate of LACV human infections that result in neurological disease. Currently, we do not know what the host factors are in humans for LACV encephalitis, whether there are strong host susceptibility factors or

whether there are factors such as where the mosquito bite occurs that could influence the potential for encephalitis.

Technique Concern

Having done intradermal inoculations on many animal species myself, I know how difficult it is to deliver a precise amount of virus. Comparative anatomy tells us that the skin of mice is thinner than other mammalian species; plus, 3-week-old weanlings (very small) were used here. For example, was there additional liquid spilling out after the inoculation? Was a needle stepper used to deliver the accurate 25 microliters of volume, and if just a regular needle was used, how was it accounted for dead volume? What gauge needle was used with what bevel? More details need to be added in the material and methods section.

We agree that we did not sufficiently provide the technical details on inoculation and have included additional information in the manuscript. All personnel inoculating mice were trained on ID inoculation and were confirmed to have consistent procedure and results with inoculation before these studies were undertaken.

Reviewer #3 (Remarks to the Author):

Overall, this work concerns a subject of great interest, often neglected, namely the role of muscle cells in arbovirus infections.

We appreciate the reviewer's comment and agree. It was very difficult finding any information on other viruses.

This study is mainly based on a mouse model only. Whereas the authors state in the introduction that La Crosse virus is of high importance in human paediatric arboviral infections, it is regrettable that no study was conducted on human primary muscle cultures, given the known differences in susceptibility to certain arboviral infections between humans and mice. In this respect, no data is available on whether in vivo or in vitro, La Crosse virus, which seems to infect differentiated muscle fibres, is able to infect myoblasts/satellite cells.

We agree with the reviewer and were able to collaborate to obtain human immortalized myoblasts to address this point (Fig. 3F).

The light/immunofluorescence microscopy images are of good quality. Unfortunately, the data produced concern very small microscopic fields, and at no time is there any mention of quantification or counting according to the usual methods in histology and anatomo-pathology. Only pictures with few LACV immunoreactive cells are provided, without any quantification.

LACV infection is a relatively rare event in the skin, but we have now included quantifications of both skin infection (Fig 1) and heart infection (Fig S1) to help address these concerns.

Pictures from mock-infected animals are lacking in some figures. Although these data provide indications on interesting new findings, they do not provide sufficient information to conclude.

We have included mock-infected animals as negative controls in Supp Fig. 2, showing the lack of virus staining.

In addition, some errors, imprecisions presented in the different figures could be corrected and improved:

In fig.1, there is confusion in the Legend: whereas in Text (Results), line 99, and Legend Fig 1 line 6, Fig 1C concerns immunofluorescence, line 9 of the legend Fig 1 suggests that it corresponds to in situ hybridization...

We have adjusted the text in this section and the figure legend to match.

In Fig 1D, how can the authors conclude that gRNA and rRNA are detected exclusively in skin-associated muscle, since no myosin labeling is mentioned?

The rRNA is in the same layer/region of the skin as we see in the fluorescence microscopy, which correlates with the muscle layer. The cells with gRNA are also the ones with rRNA—and therefore those cells with gRNA are replicating the virus. We have clarified in the text that this is indicative of cells being replication competent within the muscle layer but does not prove replication is only in the muscle.

Fig S1: it is not mentioned in the legend of the figure to which correspond the Figures S1 B' and S1B ""?? How can the authors say that "all images are representative of 2-3 mice across multiple sections"?? Does it concern 2 or 3 mice? What do the authors mean by "multiple sections? 10, 100, 1000? How can a single immunoreactive cell in a small field be representative?

We have clarified this in the figure legends to be more specific.

Fig 4B: why are there magenta labeled cells in the "anti caspase 3" column in mock?

Thank you for pointing this out. The LACV and caspase-3 images were swapped during final figure preparation (LACV is in magenta whereas the caspase-3 staining is white). We have corrected this in the revised manuscript. There is no LACV staining in the mock samples, but we do see occasional active caspase-3 positivity in the mock SMEVs.

Figure 5 showing the retention of LACV in the Golgi apparatus would have benefited from a transmission electron microscopy approach

We agree this would have been a wonderful addition to the study. We attempted TEM on these samples; however, it was very difficult to accurately pinpoint the Golgi structures by EM as they appear as punctate, highly diffuse structures within the muscle cells (see figure 5). This made it very difficult to distinguish Golgi bodies from other membranous vesicles and structures within the cell, especially as they do not cluster around the nucleus as in other cell types. In addition, the highly punctate nature of the virus throughout such long myofibers (see figure 1D) made it difficult screen sufficient sections to locate virus particles.

Figure S2: did the authors make the same experiments (testing susceptibility of isolated immune cells from the same mice towards LACV infection?)

We did not find infected immune cells in the skin during *in vivo* infection (Fig1A, CD90 staining), nor did we find them in the peritoneal muscle. Previous work in our lab failed to detect viral replication within circulating blood cells and we could not transfer virus from infected to naïve animals through PBMCs or spleen cells (Winkler et al. *Acta Neuropath* 2015). This suggests that virus does not infect and does not spread via immune cells.

REVIEWERS' COMMENTS

Reviewer #1 (Remarks to the Author):

All my concerns are addressed. This manuscript is a wonderful study. Nice work.

Reviewer #3 (Remarks to the Author):

The revised manuscript has improved in quality and content compared with the previous version. As far as I am concerned, the answers to the questions are satisfactory and the reservations expressed during the first review (incomplete or inverted legends, missing quantification, etc.) have been taken seriously into account by the authors. The additional experiments have been carried out satisfactorily; in particular, the inclusion of the study on primary human myoblasts (although they could not be differentiated into myotubes) completes the work well. Consequently, the manuscript is acceptable for publication in the journal.